

# Reconstructing climate trends adds skills to seasonal reference crop evapotranspiration forecasting

Qichun Yang[1], Quan J Wang[1], Andrew W. Western[1], Wenyan Wu[1], Yawen Shao[1], and Kirsti Hakala[1]

[1]Department of Infrastructure Engineering, The University of Melbourne, Parkville 3010, Australia

*Correspondence to*: Qichun Yang (qichun.yang@unimelb.edu.au)

**Abstract.** Evapotranspiration plays an important role in the terrestrial water cycle. Reference crop evapotranspiration ($ET_o$) has been widely used to estimate water transfer from vegetation surface to the atmosphere. Seasonal $ET_o$ forecasting provides valuable information for effective water resource management and planning. Climate forecasts from General Circulation Models (GCMs) have been increasingly used to produce seasonal $ET_o$ forecasts. Statistical calibration plays a critical role in

correcting bias and dispersion errors in $ET_o$ forecasts. However, time-dependent errors, resulting from GCM's misrepresentations of climate trends, have not been explicitly corrected in $ET_o$ forecast calibrations. We hypothesize that reconstructing climate trends through statistical calibration will add extra skills to seasonal $ET_o$ forecasts. To test this hypothesis, we calibrate raw seasonal $ET_o$ forecasts constructed with climate forecasts from the European Centre for Medium-Range Weather Forecasts (ECMWF) SEAS5 model across Australia, using the recently developed Bayesian Joint Probability

trend-aware (BJP-ti) model. Raw $ET_o$ forecasts demonstrate significant inconsistencies with observations in both magnitudes and spatial patterns of temporal trends, particularly at long lead times. The BJP-ti model effectively corrects misrepresented trends and reconstructs the observed trends in calibrated forecasts. Improving trends through statistical calibration increases the correlation coefficient between calibrated forecasts and observations (*r*) by up to 0.25 and improves the continuous ranked probability score (CRPS) skill score by up to 15% in regions where climate trends are misrepresented by raw forecasts. Skillful

$ET_o$ forecasts produced in this study could be used for streamflow forecasting, modelling of soil moisture dynamics, and irrigation water management. This investigation confirms the necessity of reconstructing climate trends in GCM-based seasonal $ET_o$ forecasts, and provides an effective tool for addressing this need. We anticipate that future GCM-based seasonal $ET_o$ forecasting will benefit from correcting time-dependent errors through trend reconstruction.

## 1 Introduction

As a critical process in the terrestrial water cycle, evapotranspiration transfers a large amount of water from the land surface to the atmosphere. Reference crop evapotranspiration ($ET_o$) measures the evaporative demand of the atmosphere and thus provides valuable information for understanding and simulating terrestrial hydrology. Forecasting of $ET_o$ has been used to support water resource management (Anderson et al., 2015; Le Page et al., 2021) and improve soil moisture modelling (Yu et al., 2016). In addition, $ET_o$ forecasting also helps constrain the significant uncertainties in streamflow forecasting (Greuell et





al., 2019; Van Osnabrugge et al., 2019). Seasonal $ET_o$ forecasts have been used to support water allocation among competing
users (Chauhan and Shrivastava, 2009) and in planning farming activities (Zinyengere et al., 2011). In recent years, climate
forecasts produced by General Circulation Models (GCMs) have been adopted for seasonal $ET_o$ forecasting, since GCMs often
produce forecasts of all climate variables needed to estimate future $ET_o$ (Tian et al., 2014; Zhao et al., 2019a).

Raw $ET_o$ forecasts constructed with GCM climate forecasts often inherit significant errors from the raw forecasts of climate
variables, including temperature, solar radiation, wind speed, and vapor pressure. Due to deficiencies in GCM's representation
of physical processes of the atmosphere (Woldemeskel et al., 2014), model parameterization (O'Gorman and Dwyer, 2018),
and data assimilation (O'kane et al., 2019), raw GCM forecasts often demonstrate systematic errors (Weisheimer and Palmer,
2014). For example, inconsistencies with observations have been reported for the raw forecasts of all variables needed to
construct $ET_o$ forecasts using the Food and Agriculture Organization (FAO) 56 method (Groisman et al., 2000; Slater et al.,
2017). These inconsistencies often lead to significant bias and low skills in the resultant raw $ET_o$ forecasts (Zhao et al., 2019b).
Failing to correctly simulate the temporal trends of the climate system could be partially responsible for the low skills of GCM-
based raw $ET_o$ forecasts. Time-dependent errors are introduced when GCMs lack skills in modelling climate trends driven by
rising atmospheric greenhouse gas (GHG) concentrations (Sansom et al., 2016). There is mounting evidence that climate
change has resulted in increasing trends in temperature (Smith et al., 2007) and vapor pressure (Byrne and Gorman, 2018), but
led to decreasing trends in solar radiation (Liepert, 2002). However, GCMs configured for seasonal climate forecasts often
misrepresent these observed trends. For example, an evaluation across nine climate regions in the U.S. showed that nine of ten
selected GCMs failed to reproduce the observed temporal trends in seasonal temperature forecasts (Bhowmik and
Sankarasubramanian, 2020). In the Middle East, seasonal temperature forecasts by the Climate Forecast System version 2
(CFSv2) model overestimated the warming trend in reference data by approximately $0.4°$ decade$^{-1}$ (Alizadeh-Choobari et al.,
2019). In Australia, evaluations of the European Centre for Medium-Range Weather Forecasts (ECMWF) SEAS5 model
demonstrated significant discrepancies between observed and forecasted trends in temperature (Shao et al., 2020, 2021).
Forecasts of fire weather index (calculated with forecasts of precipitation, wind speed, temperature, and humidity) based on
the ECMWF System 4 model demonstrated significant inconsistencies with observations in temporal trends in Europe during
1981-2010 (Bedia et al., 2018). As a result, it is unlikely that raw $ET_o$ forecasts constructed with raw forecasts of these climate
variables would faithfully reproduce the observed climate trends. Failing to capture the observed trends inevitably introduces
errors to GCM-based raw $ET_o$ forecasts.

Raw $ET_o$ forecasts constructed with climate forecasts need to be calibrated to correct biases and dispersion errors. Statistical
calibration models initially developed for other variables, such as precipitation or temperature, have been adopted to calibrate
raw $ET_o$ forecasts (Medina and Tian, 2020; Zhao et al., 2019a). Using a quantile-mapping method, Tian and Martinez (2014)
improved seasonal $ET_o$ forecasts based on CFSv2 outputs in Florida, the U.S. In the calibration of seasonal $ET_o$ forecasts in
Australia, Zhao et al. (2019b) used the Bayesian Joint Probability (BJP) model to post-process $ET_o$ forecasts constructed with
climate forecasts from the Australian Bureau of Meteorology's Australian Community Climate and Earth-System Simulator-
Seasonal prediction system version 1 (ACCESS-S1) model across three weather stations. This investigation validated the BJP

model's strengths in error correction and skill enhancement in $ET_o$ forecasting. However, none of these calibrations have

explicitly dealt with time-dependent errors caused by the misrepresentation of climate trends in GCM forecasts.

Statistical techniques have been developed to correct time-dependent errors in raw GCM forecasts. A commonly adopted method is to replace the linear trend in raw forecasts with the observed trend (Kharin et al., 2012). Using this method, Kharin et al. (2012) corrected trends in decadal temperature forecasts and successfully reduced the systematic residual drifts in raw forecasts. Meanwhile, improvements in trends effectively adjusted the long-term climate behavior in forecasts to match

observations (Kharin et al., 2012). To correct errors associated with the representation of temporal changes and variability, Pasternack et al. (2020) adopted a time-varying mean to characterize the climate trend in the calibration of decadal temperature forecasts. In addition to these decadal-scale calibrations, recent studies suggested that seasonal climate forecasting could also benefit from correcting time-dependent errors. For example, Shao et al. (2021) improved the BJP model by adding trend-reconstruction algorithms to deal with time-dependent errors. The new algorithm allows for the reconstruction of observed

trends in calibrated forecasts. With this new feature, the improved BJP model (hereafter referred to as BJP-ti) demonstrates the capability of adding extra skills to seasonal temperature forecasts through reconstructing observed trends in calibrated forecasts.

We hypothesize that reconstructing trends in seasonal $ET_o$ forecasts through statistical calibration will help correct time-dependent errors and thereby improve forecast skills. To test this hypothesis, we adopt the BJP-ti model to calibrate seasonal

$ET_o$ forecasts constructed with climate forecasts from the ECMWF SEAS5 model across Australia. This investigation aims to 1) reconstruct climate trends in seasonal $ET_o$ forecasts through statistical calibration and 2) investigate how trend reconstruction affects the skill of calibrated $ET_o$ forecasts.

## 2 Method

### 2.1 Observations and forecasts

We develop monthly $ET_o$ data (treated as observations for calibration) based on gridded monthly temperature, solar radiation, and vapor pressure data from the Australian Water Availability Project (AWAP) (Jones et al., 2007, 2014). Since the AWAP project does not provide wind speed data, we use a constant wind speed of 2 m s$^{-1}$ in deriving the $ET_o$ observations (Allen et al., 1998). Based on these AWAP variables, we produce monthly $ET_o$ observations during 1990-2019 for forecast calibration. Seasonal climate forecasts from the latest version (SEAS5) of the ECMWF model are used to construct the raw $ET_o$ forecasts.

The re-forecast period of SEAS5 is 1981–2016, and the ensemble size is 25 members. SEAS5 forecasts have a horizon of seven months (months 0 to 6), with a spatial resolution of 0.4°. Real-time forecasts started in 2017, with an ensemble size of 51 members (Stockdale et al., 2017). While SEAS5 produces climate forecasts across the globe, the calibration in this study is performed across Australia only.

To match $ET_o$ observations, we combine the archived re-forecasts and operational forecasts to derive raw $ET_o$ forecasts for

the period of 1990-2019. We choose the first 25 ensemble members of the real-time forecasts (2017-2019) to match the





ensemble size of the re-forecasts (1990-2016). Next, we calculate the ensemble mean of the 25 ensemble members of ECMWF forecasts of temperature, solar radiation, and vapor pressure for the calculation of raw $ET_o$ forecasts. To be consistent with the $ET_o$ observations, we use a constant wind speed of 2 m s$^{-1}$ in deriving raw $ET_o$ forecasts. Finally, we aggregate the grid spacing of AWAP data from 0.05° to match the ECMWF's spatial resolution of 0.4°.

**2.2 Calculation of $ET_o$**

We construct monthly raw $ET_o$ forecasts and $ET_o$ observations using the monthly ECMWF climate forecasts and AWAP data based on the FAO 56 $ET_o$ method (Allen, et al., 1998):

$$ET_O = \frac{0.408\Delta(R_n-G)+\gamma\frac{900}{T+273}u_2(e_s-e_a)}{\Delta+\gamma(1+0.34u_2)} \tag{1}$$

where $ET_O$ is the monthly reference crop evapotranspiration ($mm\ month^{-1}$); $\Delta$ is the slope of the vapor pressure curve

($kPa\ °C^{-1}$); $R_n$ is net radiation at the crop surface ($MJ\ m^{-2}\ month^{-1}$); $G$ is soil heat flux density ($MJ\ m^{-2}\ month^{-1}$), which is calculated based on temperature; $\gamma$ is the psychrometric constant ($kPa\ °C^{-1}$); $T$ is average air temperature ($°C$); $u_2$ is the wind speed at 2 m ($m\ s^{-1}$); and $e_s$ and $e_a$ are saturated and actual vapor pressure ($kPa$), respectively.

**2.3 Forecast calibration with the BJP-ti model**

In this study, $ET_o$ forecast calibration is conducted across Australia for each grid cell, each month, and lead time separately

during 1990-2019. We employ the BJP-ti model to calibrate the raw $ET_o$ forecasts. This model was developed recently by extending the original BJP model's capability to deal with errors resulting from the misrepresentation of climate trends. In this study, the calibration model is configured by month k (k = 1 to 12 corresponding to January to December) of the year.
Calibration with the BJP-ti model involves six steps, including 1) data transformation, 2) data detrending, 3) joint probability modelling of the transformed, detrended forecasts and observations, 4) generation of ensemble calibrated forecast members

conditional on the raw forecast, 5) adding the observed trend back to ensemble members, and 6) back-transforming the data to obtain the final calibrated forecasts. We further introduce these steps in detail as follows.
The first calibration step is to transform raw forecasts and observations to approach the normal distribution. We adopt the Yeo-Johnson transformation method (Yeo and Johnson, 2000) to transform $ET_o$:.

$$x' = \begin{cases} (\lambda x + 1)^{\frac{1}{\lambda}} - 1, & (x \geq 0, \lambda \neq 0) \\ exp(x) - 1, & (x \geq 0, \lambda = 0) \end{cases} \tag{2}$$

where $\lambda$ is a transformation parameter; $x$ refers to raw $ET_o$ forecasts or $ET_o$ observations ($mm\ month^{-1}$); $x'$ is the transformed $x$ (forecasts or observations) generated through the Yeo-Johnson transformation. The above transformation is performed by month of the year for raw forecasts and observations separately. The transformation parameter ($\lambda$) is inferred using the Bayesian Maximum a Posterior (MAP) method (Shao et al., 2020).



Step 2 is to generate detrended forecasts and observations in the transformed space. For each grid cell, we infer linear trends

for transformed forecasts and observations separately. With the trend parameters ($\alpha_f$ and $\alpha_o$), trends in transformed forecasts

and observations are removed to produce detrended data. Specifically, each transformed forecast and observation record is

adjusted based on the middle year of the study period (1990-2019) and trend parameters using the following equations:

$$z_f(t) = y_f'(t) - \alpha_f(t - t_m) \tag{3}$$

$$z_o(t) = y_o'(t) - \alpha_o(t - t_m) \tag{4}$$

where $y_f'(t)$ and $y_o'(t)$ refer to transformed $ET_o$ forecasts and observations for month $k$ ($k$ = 1 to 12 corresponding to January

to December) in year $t$ of 1990-2019; $\alpha_f$ and $\alpha_o$ are inferred trend parameters for transformed forecasts and observations,

respectively; $t_m$ is approximately the middle year (e.g., 2004 in this study) during 1990-2019; and $z_f(t)$ and $z_o(t)$ are

detrended $ET_o$ forecasts and observations in the transformed space, respectively.

In step 3, we assume a bivariate joint distribution ($z$) between predictor $z_f$ (detrended transformed raw forecasts) and predictand

$z_o$ (detrended transformed observations)

$$z = \begin{bmatrix} z_f \\ z_o \end{bmatrix} \sim N(\mu, \Sigma) \tag{5}$$

where μ is the mean vector, and Σ is the covariance matrix. We denote the parameters from equations 3-5 as a vector $\theta = \{\mu, \Sigma, \alpha_f, \alpha_o\}$.

For each month of the year, model parameters are inferred with training data pairs (predictor and predictand) during the study

period (1990-2019). The posterior distribution of the model parameters is:

$$p(\theta|D) \propto p(\theta)p(D|\theta) = p(\theta) \prod_{t=1}^{n} p(D|\theta) \tag{6}$$

where $p(\theta)$ is the prior distribution for model parameters, and $p(D|\theta)$ is the likelihood function. D refers to all data pairs

($z_f(t)$ and $z_o(t)$) used for parameter inference. A Gibbs sampler is utilized to repeatedly sample the parameter sets θ from the

conditional posterior distribution of the model parameters.

In the BJP-ti model, informative priors are applied to set boundaries for inferred trends to avoid extreme values for each grid

cell, month, and lead time. This informative prior distribution $p(\alpha_i)$ for trend parameters $\alpha_f$ and $\alpha_o$ is formulated as follows

(Shao et al., 2021):

$$p(\alpha_i) \propto N(0, m_i^2) \tag{7}$$

$$[\alpha_i | \cdot] = N\left(\frac{m_i^2 \sum_{t=1}^{n}(y_i'(t) - \mu_i)(t - t_m)}{m_i^2 \sum_{t=1}^{n}(t - t_m)^2 + \sigma_i^2}, \frac{m_i^2 \sigma_i^2}{m_i^2 \sum_{t=1}^{n}(t - t_m)^2 + \sigma_i^2}\right) \tag{8}$$

where $m_i$ is the standard deviation of the prior, which is set based on trends of transformed forecasts and observations. To

determine $m_i$, we pooled the trends of all grid cells, months, and lead times for transformed forecasts, and found that 95% of

the absolute trends are smaller than 0.43. For transformed observations, 95% of grid cells and months have absolute trends

less than 0.49. As a result, we set $m_i$ to 0.43 and 0.49 for forecasts and observations, respectively. Equation 8 shows the





conditional posterior distribution of parameter $\alpha_i$. In equation 8, $\mu_i$ is the mean and $\sigma_i$ is the standard deviation for predictors

or predictands.

In step 4, once all the parameters are inferred, we draw 1000 members from a conditional distribution of the predictand $(z_o(t^*))$, for a given new forecast $(z_f(t^*))$. In step 5, we add the trend from Equation 4 back to $z_o(t^*)$, to produce calibrated ensemble forecast $(y'_o(t^*))$. In step 6, we back-transform $y'_o(t^*)$ to the original space to produce the calibrated ensemble forecasts.

**2.4 Evaluation of forecast calibration**

To evaluate the performance of the calibration, we adopt a leave-one-year-out cross-validation strategy for each grid cell and lead time. Specifically, for one of the 30 years during 1990-2019, we keep month k aside, and then use month k from the remaining 29 years to infer the BJP-ti parameters. Once the parameters are inferred, we generate a calibrated forecast for month k in the year held aside. This process is repeated until a calibrated forecast is obtained for month k from each of the 30 years. Similar processes are conducted for other months and other lead times until we obtain calibrated forecasts for all months and

the seven lead times for each cell across Australia.

To evaluate how the reconstruction of trends affects the quality of calibrated forecasts, we compare BJP-ti calibrated forecasts with those generated using the original BJP model, which does not reconstruct trends. The BJP model omits steps 2 (detrending) and 5 (retrending) in section 2.3. The comparison is conducted for months with large areas of statistically significant (at the 95% confidence interval) temporal trends in observed ET$_o$.

Evaluation metrics employed to examine the performance of calibrations include correlation coefficients, skill score, bias, and reliability. The calculation of these metrics is further introduced as follows.

**2.4.1 Correlation coefficient**

We use the Pearson correlation coefficient ($r$) between raw/calibrated forecasts and observations in each month to examine their consistency in temporal dynamics:

$$r = \frac{\sum_{t=1}^{T}(x(t)-\bar{x})(y(t)-\bar{y})}{\sqrt{\sum_{t=1}^{n}(x(t)-\bar{x})^2}\sqrt{\sum_{t=1}^{n}(y(t)-\bar{y})^2}} \qquad (9)$$

where $x(t)$ is the ensemble mean of raw/calibrated ET$_o$ forecasts for month $k$ in year $t$ (*mm month$^{-1}$*); $T$ is the total years during the study period; $\bar{x}$ is the average of $x(t)$ (*mm month$^{-1}$*); $y(t)$ is the corresponding ET$_o$ observations of the same month (*mm month$^{-1}$*), and $\bar{y}$ is the average of $y(t)$ (*mm month$^{-1}$*).

**2.4.2 Forecast skills**

We use the continuous ranked probability score (CRPS) to measure the skill of the raw and calibrated forecasts (Grimit et al., 2006):

$$CRPS(t) = \int \{F(t,x) - H(x - y(t))\}^2 dx \qquad (10)$$

$$\overline{CRPS} = \frac{1}{n}\sum_{t=1}^{n} CRPS(t) \qquad (11)$$





where $F(t, x)$ is the cumulative density function of an ensemble forecast, and $y(t)$ is the observation at time $t$; $H$ is the
Heaviside step function ($H = 1$ if $x - y(t) \geq 0$ and $H = 0$ otherwise); the overbar represents averaging across the $n$ months
during 1/1990-12/2019. For deterministic raw forecasts, CRPS is reduced to absolute errors.

We further calculate the CRPS skill score ($CRPS_{SS}$) to measure the skill of raw and calibrated forecasts relative to climatology
forecasts using the following equation:

$$CRPS_{SS} = \frac{CRPS_{reference} - CRPS_{forecasts}}{CRPS_{reference}} \times 100 \qquad (12)$$

where $CRPS_{reference}$ is the CRPS value of climatology forecasts; and $CRPS_{forecasts}$ refers to CRPS value of raw or calibrated
forecasts. Positive $CRPS_{SS}$ indicates better skill than the climatology forecasts, and vice versa. To make the CRPS skill scores
of calibrated forecasts generated by different models (BJP vs. BJP-ti) comparable, we use the climatology forecasts from the
BJP model as the reference in the calculation of $CRPS_{SS}$.

### 2.4.3 Bias

We evaluate the accuracy of the raw and calibrated forecasts using the following equation:

$$Bias = \frac{1}{n}\sum_{t=1}^{n}(x(t) - y(t)) \qquad (13)$$

where $Bias$ refers to the bias in ET$_o$ (*mm month$^{-1}$*); $n$ is total months during the 30-year study period (1/1990-12/2019); $x(t)$
is raw or calibrated forecasts of ET$_o$ (*mm month$^{-1}$*), and $y(t)$ is the corresponding ET$_o$ observations of the same month (*mm
month$^{-1}$*).

### 200 2.4.4 Reliability

To evaluate the reliability of calibrated ensemble forecasts, we calculate the probability integral transform (PIT) value using
the following equation:

$$\pi(t) = F(t, x = y(t)) \qquad (14)$$

where $F(t, x)$ is the cumulative density function of the ensemble forecast, and $y(t)$ is the observation. For reliable forecasts,
the collection of $\pi(t)$ follows a standard uniform distribution. We use the alpha ($\alpha$) index to summarize the reliability in each
grid cell with the following equation to check the overall reliability across Australia (Renard et al., 2010):

$$\alpha = 1 - \frac{2}{n}\sum_{t=1}^{n}\left|\pi^*(t) - \frac{t}{n+1}\right| \qquad (15)$$

where $\pi^*(t)$ is the sorted $\pi(t)$, t=1,2,…n in ascending order, and $n$ is the total number of months. The $\alpha$-index measures the
total deviation of calibrated forecasts from the corresponding uniform quantile. Perfectly reliable forecasts should have an $\alpha$-
index of 1, and forecasts with no reliability would have an $\alpha$-index of 0.

### 3 Results





## 3.1 Trends in observations and raw/calibrated forecasts

We evaluate the capability of BJP-ti in reconstructing temporal trends for months with large areas of statistically significant trends in observed $ET_o$. Since the trend parameters are estimated by month, we first examine the trend in $ET_o$ observations for

each month $k$ of the year for 1990-2019 (Figure S1). August, September, and October show larger areas with statistically significant trends than other months. As a result, the evaluation of trends in raw/calibrated forecasts is mainly conducted for these three months.

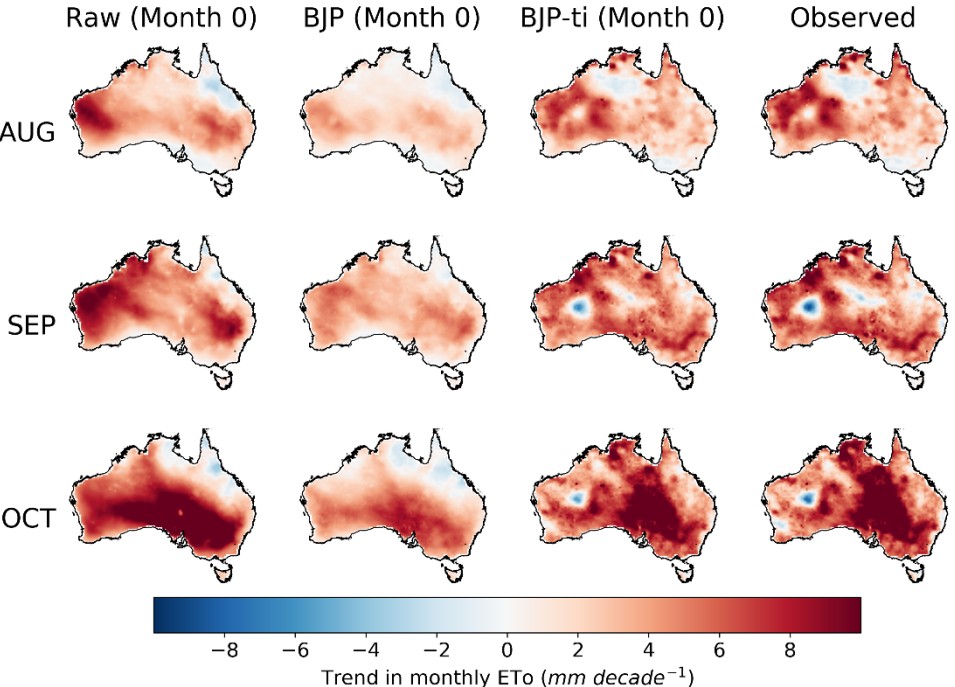

**Figure 1. Trends in raw forecasts, BJP calibrated forecasts, and BJP-ti calibrated forecasts at the lead time of month**

**0, and observed $ET_o$ in August, September, and October.**

Observed $ET_o$ shows increasing trends in many parts of Australia in the three selected months (Figure 1, right column). In August, areas with increasing trends larger than 6 *mm decade^-1* are mainly located in western parts of Australia. In contrast, central and eastern Australia demonstrates much lower trends of less than 4 *mm decade^-1*. Observed trends are close to zero in Victoria and Tasmania and even become negative in parts of the Northern Territory. In September, areas with significant

increasing trends larger than 6 *mm decade^-1* are located in many parts of Australia, with the exception of a narrow coastal fringe and areas around the Tropic of Capricorn. In this month, decreasing trends are observed in a small part of eastern areas of Western Australia, where observations are relatively poor. In October, central-eastern Australia, including the inland regions of Victoria, New South Wales, South Australia, and south-west Queensland, demonstrate increasing trends of up to 8 *mm decade^-1*.



Raw $ET_o$ forecasts also demonstrate trends, but they differ from those in observations in both spatial patterns and magnitudes (left column in Figure 1). In August, raw forecasts show increasing trends (> 6 *mm decade$^{-1}$*) in Western Australia, which partially match those in observations. However, in eastern parts of Australia, raw forecasts overpredict trends in observations. In September, raw forecasts demonstrate even larger overpredictions (>8 *mm decade$^{-1}$*) in trends than those of August, particularly in Western Australia and New South Wales. In October, raw forecasts are better aligned with observations in the

increasing trends in south-eastern Australia; however, they overpredict trends in Western Australia, and underpredict trends in Northern Australia.

   Trends in raw forecasts become weaker at longer lead times (left columns in Figures S2 and S3). For the lead time of month 3, trends in raw $ET_o$ forecasts show similar spatial patterns to those of month 0 in August, but the trends mainly drop to less than 2 *mm decade$^{-1}$*. Similarly, the magnitudes of increasing trends in the other two months are also much lower at month 3 than those at month 0.

At month 6, trends in raw forecasts of the three selected months are close to zero across Australia.

   Calibrated $ET_o$ forecasts produced with the original BJP model demonstrate trends similar to those of raw forecasts in spatial patterns, but show smaller magnitudes (second columns in Figures 1, S2, and S3). Specifically, at month 0, the BJP-calibrated forecasts preserve the spatial variability of trends of the raw forecasts and show higher trends in Western Australia, central parts of Australia, and southern regions of the country for August, September, and October, respectively, but the increasing

trends are all less than 4 *mm decade$^{-1}$*, lower than those in raw forecasts (Figure1). Consistencies in the spatial patterns of trends are also found between BJP-calibrated forecasts and raw forecasts at other lead times (Figures S2 and S3). Similarly, trends are also lower in BJP-calibrated forecasts than those of the corresponding raw forecasts at longer lead times.

   Calibration with the BJP-ti model successfully reconstructs the observed trends in the calibrated forecasts (third columns in Figures 1, S2, and S3). Inconsistencies between raw forecasts and observations in the spatial patterns and magnitudes of trends

are effectively corrected through statistical calibration. In addition, the tendency that trends become weaker at longer lead times in the raw forecasts is also effectively corrected. In the BJP-ti calibrated forecasts (third columns in Figures 1, S2, and S3), all lead times show trends consistent with observations in both spatial patterns and magnitudes.

**3.2 Correlation coefficients between forecasts and observations**





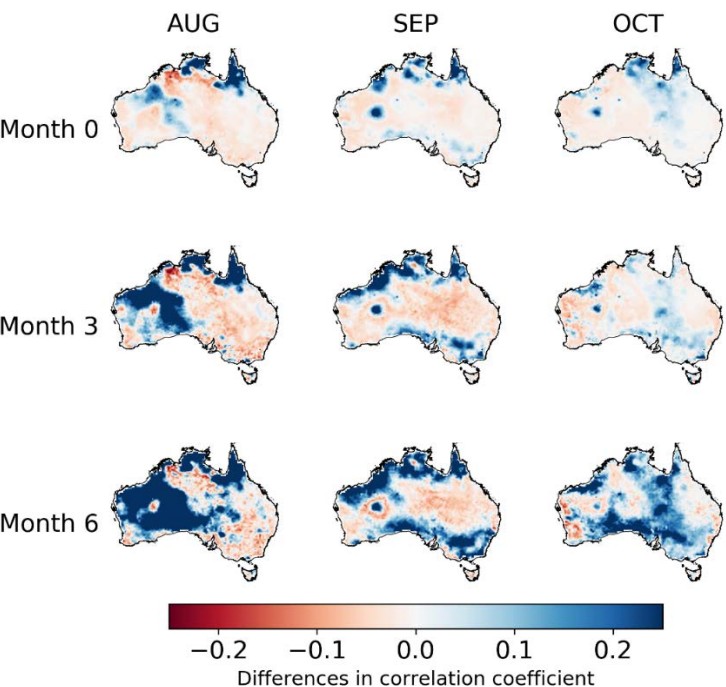

Fig. 2. Differences in the correlation coefficient (*r*) between BJP-ti calibrated forecasts and observations with that between BJP calibrated forecasts and observations for three selected months (AUG, SEP, OCT) and three lead times (months 0, 3, and 6)

We further examine whether reconstructing trends improves the representation of $ET_o$ temporal dynamics by forecasts. Specifically, we compare the *r* between BJP-ti calibrated forecasts and $ET_o$ observations with that between BJP-calibrated forecasts and observations in August, September, and October (Figure 2). Following trend reconstruction, BJP-ti calibrated forecasts clearly present temporal patterns more consistent with observations than calibrated forecasts produced by the BJP model, particularly in regions where observations show significant trends (Figure S1), and for forecasts at longe lead times. For the lead time of month 0, increases in *r* of over 0.1 are mainly located in the coastal regions of Northern Australia and northern Queensland for all the three selected months. More significant improvements in *r* are found at longer lead times (3 and 6 months), with larger areas showing increases in *r* (Figure 2). At month 3, in addition to the coastal areas in northern Australia, the majority of Western Australia shows increases in *r* by more than 0.2 in August; in September, significant increases in *r* occur in both the far north and far south of mainland Australia; in October, areas with higher *r* further expand in southern Australia, and cover much larger areas than those at month 0. Areas showing higher *r* continue to expand at month 6. In August, increases in *r* of over 0.2 or even 0.25 are found in western and central far northern Australia; in September, regions with higher *r* cover large areas in coastal parts of northern Australia and coastal regions across Victoria and South Australia. In October, *r* increases cover large areas of southern and central regions of Australia. Slight decreases in *r* are also found in regions where the observed trends are not statistically significant.



### 3.3 Skills of raw and calibrated ET$_0$ forecasts

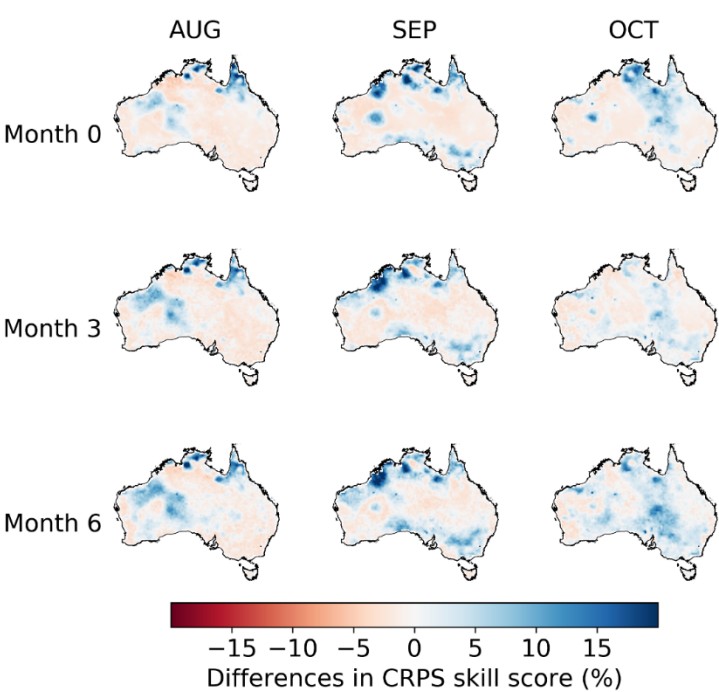

**Figure 3. Differences in CRPS skill score between BJP-ti calibrated forecasts and the BJP calibrated forecasts for three selected months (AUG, SEP, OCT) and three lead times (months 0, 3, and 6)**

Reconstruction of trends results in more skillful calibrated forecasts. We compare the CRPS skill scores of BJP-ti calibrated forecasts with those produced with the BJP model for the three selected months (Figure 3). At month 0, the CRPS skill score of calibrated forecasts is increased by 5-10% in August, September, and October, when trends are reconstructed. The distribution of areas with increased CRPS skill scores is generally consistent with that of the improved $r$ (Figure 2). Increases in CRPS skill score are greater at longer lead times, in both magnitude and area, than those at short lead times. At month 3, areas with increased CRPS skill scores expand in Western Australia in August and in northern Western Australia in September. Month 6 demonstrates further improvements, with larger areas showing increases in CRPS skill score of over 15% in coastal areas of northern Australia in August and September, and central Australia in October. Slight decreases in CRPS skill score are also found in regions where the observed trends are not statistically significant.





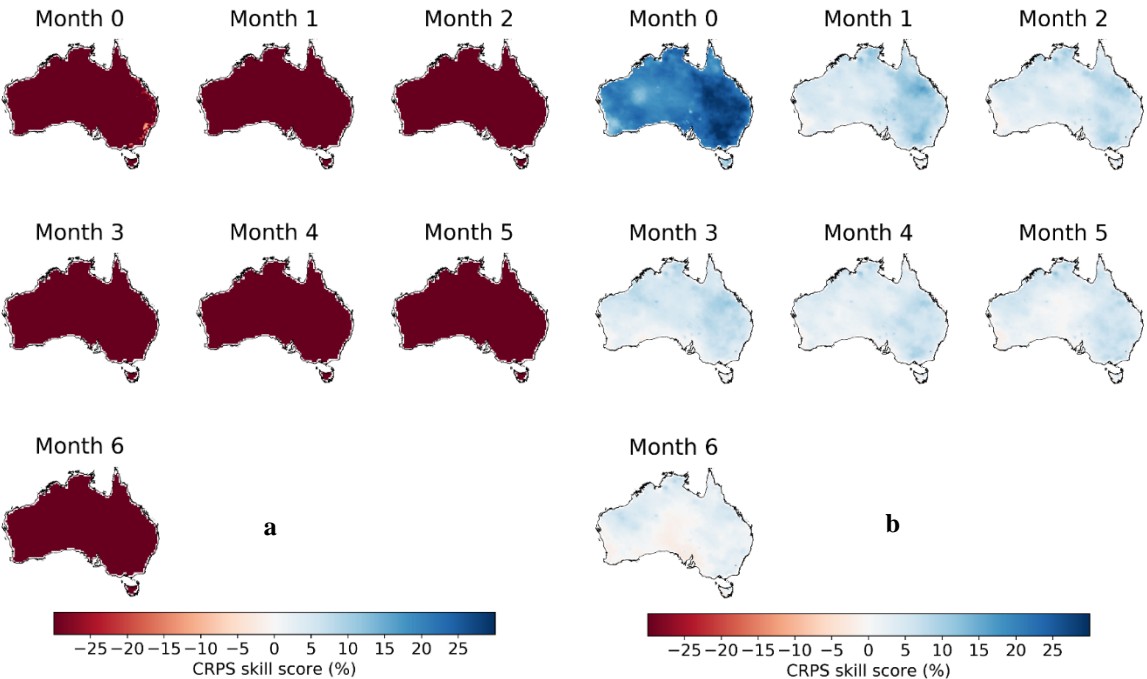

**Figure 4. CRPS skill score in (a) raw and (b) calibrated forecasts at seven lead times during 1990-2019.**

We further evaluate the overall performance of the calibration over the whole study period by comparing CRPS skill scores of the raw and BJP-ti calibrated forecasts (Figure 4). Calibration with BJP-ti substantially improves the skills of the raw $ET_o$

forecasts. Compared with the climatology forecasts, raw $ET_o$ forecasts demonstrate much lower skills, with CRPS skill scores lower than -25% in all grid cells, even for those at short lead times. With the correction of errors, including the time-dependent errors, the BJP-ti calibrated forecasts demonstrate CRPS skill scores more significant than 20% at month 0 in most grid cells. Eastern parts of Australia, such as New South Wales and Victoria, show CRPS skill scores of up to 30%. Beyond month 0, the skill score decreases significantly in calibrated forecasts. Most areas of Australia show CRPS skill scores lower than 10% at

month 1. The skill score further decreases at longer lead times, but remains above zero in many parts of Australia, even at month 6, suggesting better performances than the climatology forecasts.





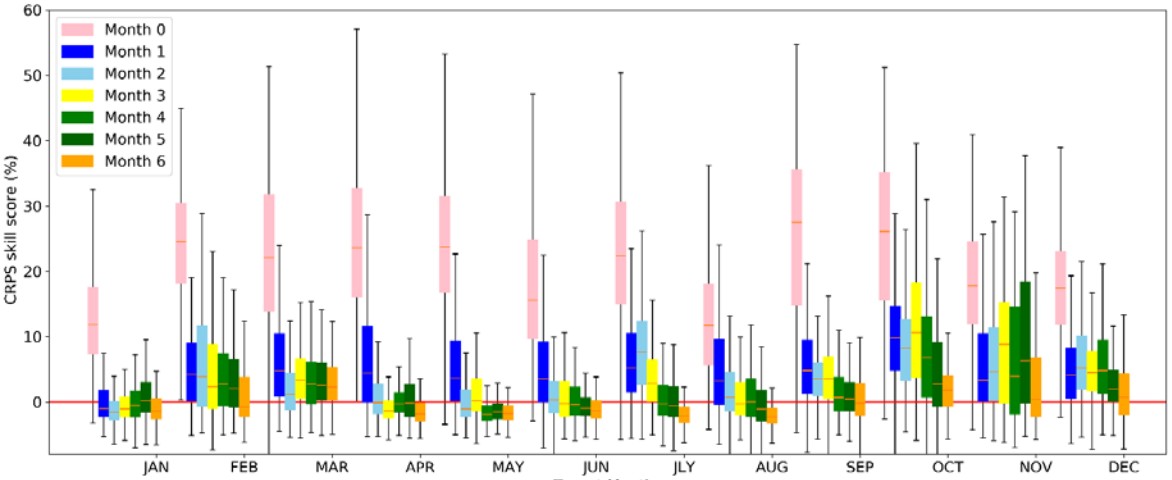

**Figure 5. Boxplot of CRPS skill score by target month in BJP-ti calibrated forecasts**

We also summarize the CRPS skill score of calibrated forecasts by target month at the seven lead times across Australia (Figure

5). Individual boxes indicate the variability among all the grid cells across Australia for that month and lead-time. At the first

lead time (month 0), all months show CRPS skill score markedly better than climatology forecasts across most grid cells, with

the median CRPS skill score being above 20% for seven months. However, the skill score decreases quickly with lead time.

At lead time 1, the CRPS skill score is mainly lower than 10% for all target months. Skills of calibrated forecasts vary among

the months. For October, November, and December, the CRPS skill score is above 0 for more than 50% of grid cells, even at

lead time 6, indicating better performance than the climatology forecasts. For other months, such as January, April, May, and

June, the median CRPS skill score decreases to values slightly below 0 beyond the lead time of month 1.

**3.4 Bias in raw and BJP-ti calibrated ET$_0$ forecasts**


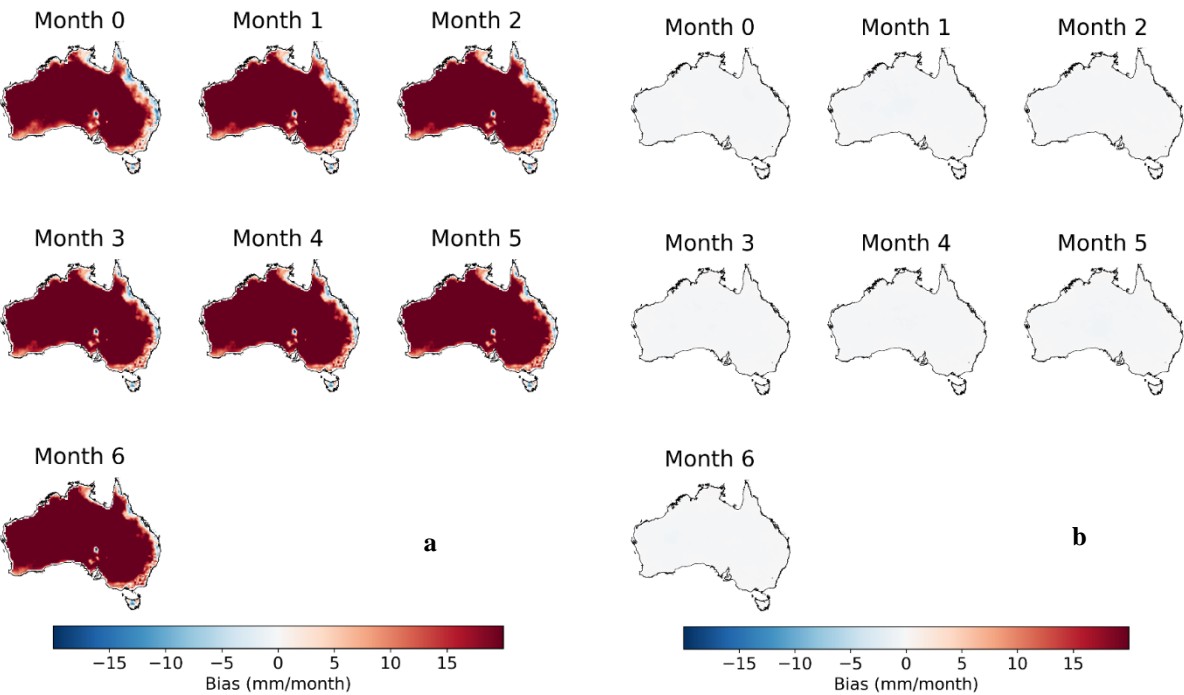

**Figure 6. Bias in (a) raw and (b) BJP-ti calibrated ET$_o$ forecasts at 7 lead times during 1990-2019 across Australia**

Raw monthly ET$_o$ forecasts constructed with the raw climate forecasts of the ECMWF SEAS5 model demonstrate significant overpredictions (Figure 6). Positive biases of over 15 *mm month$^{-1}$* occur in most parts of Australia, away from the coastal fringe and Tasmania (Figure 6). Small areas with negative biases are found in the coastal margins of Queensland and Tasmania. The spatial patterns of bias in the raw ET$_o$ forecasts are consistent across all seven lead times, demonstrating systemic errors in raw ET$_o$ forecasts (Figure 6). The BJP-ti calibration substantially corrects the systematic errors in the raw forecasts, resulting in biases close to 0 in calibrated forecasts for all lead times (Figures 6 and S4).

### 3.5 Reliability of calibrated ET$_o$ forecasts





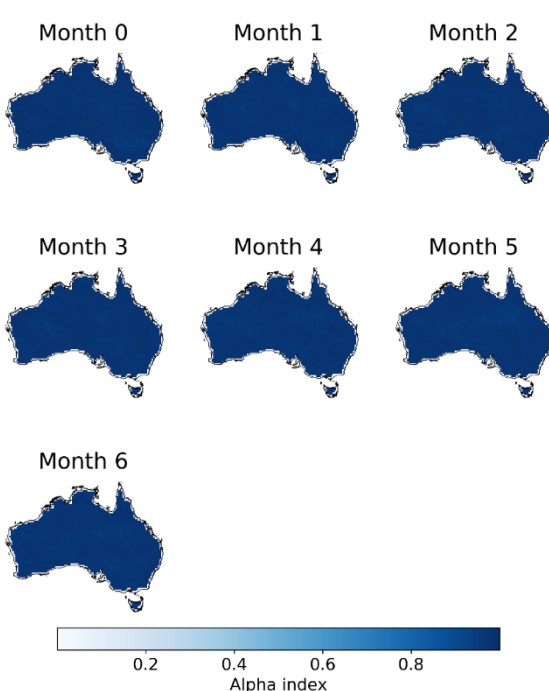

**Figure 7. Alpha index of BJP-ti calibrated ensemble $ET_o$ forecasts**

In this study, we generate 1000 ensemble members for each raw forecast to quantify the uncertainties of the calibrated forecasts.

As indicated by the α-index, calibrated $ET_o$ forecasts are highly reliable. The α-index of calibrated ensemble $ET_o$ forecasts is above 0.96 in most parts of Australia for all the seven lead times (Figures 7 and S5). The high reliability of the calibrated forecasts suggests reasonable representations of uncertainties in calibrated $ET_o$ forecasts, and the distributions of calibrated ensemble forecasts are neither too narrow nor too wide (Figure 7).

## 4 Discussion

**4.1 The necessity of reconstructing climate trends in seasonal $ET_o$ forecasting**

This investigation confirms that the misrepresentation of climate trends is an important error source in GCM-based $ET_o$ forecasting. Most previous investigations on climate trends in seasonal forecasts were primarily focused on temperature (Krakauer, 2019) and precipitation (Alizadeh-Choobari et al., 2019), and existing $ET_o$ forecasting studies have not investigated trends in $ET_o$ forecasts, despite temporal trends in $ET_o$ being observed at weather stations across the globe (Djaman et al.,

2018; Kousari and Ahani, 2012). Although the ECMWF model runs have been forced with the observed greenhouse gas concentrations for our study period (Johnson et al., 2019), and have actually produced temporal trends in raw $ET_o$ forecasts (Figure 1), the trends show significant inconsistencies with observations. In addition, raw $ET_o$ forecasts at long lead times





demonstrate much weaker trends than those at short lead times. Since misrepresentations of climate trends have been reported for many GCMs (Dunn et al., 2017), GCM-based seasonal $ET_o$ forecasting may generally suffer from time-dependent errors.

This investigation also verifies our hypothesis that correcting time-dependent errors through trend reconstruction can add extra skills to calibrated $ET_o$ forecasts. Reconstruction of climate trends using the BJP-ti model effectively improves the consistency between forecasts and observations in temporal patterns and leads to more skillful calibrated forecasts, when compared with the calibration that does not reconstruct trends in $ET_o$ forecasts. These improvements are particularly significant in regions showing statistically significant observed trends, and at long lead times when trends are misrepresented most. Consequently,

this investigation clearly indicates the necessity of correcting time-dependent errors in seasonal $ET_o$ forecasting. We recommend that future GCM-based $ET_o$ forecasting should correct time-dependent errors, since climate change has been projected to intensify in the future (Kharin et al., 2013), and may induce more significant temporal trends in $ET_o$.

### 4.2 Implications for improving statistical calibration models

Climate change has posed challenges to the statistical calibration of seasonal climate forecasts. Many post-processing models,

such as those based on the probabilistic theory (Tian et al., 2014; Wang et al., 2009), often rely on the climatology of observations to construct the probability distribution function for calibration (Wilks, 2018). However, the non-stationary behavior of the climate system induced by elevated greenhouse gas emissions has been increasingly reported (Haustein et al., 2016; Lima et al., 2015). Many calibration models developed for seasonal forecasts have not considered the climate change impacts on the observed climatology. Although these models are proven to be effective in correcting biases in raw forecasts,

assuming a static climatology may have hindered the utilization of predictable information in the raw forecasts. This investigation and our previous calibration of seasonal temperature forecasts (Shao et al., 2020, 2021), suggest that reconstructing trends in calibrated forecasts is an effective solution for capturing the non-stationary behavior of the climate system for more robust statistical calibrations of seasonal climate forecasts.

This current investigation has further validated the strength of the trend-reconstruction algorithms in BJP-ti. Previously, we

applied this model to correct seasonal temperature forecasts and achieved significant improvements in forecast skills relative to the original BJP model (Shao et al., 2020, 2021). The successful application to $ET_o$ forecasts confirms the robustness of trend reconstruction algorithms based on the data transformation and Bayesian inference in BJP-ti. This study further demonstrates the feasibility for the general application of BJP-ti to different hydroclimate variables showing temporal trends. We also anticipate that the BJP-ti algorithms for trend reconstruction could be adopted by other calibration models to enhance

seasonal forecast calibration.

### 4.3 Future work for seasonal $ET_o$ forecasting

In this investigation, we successfully improve $ET_o$ forecast calibration by reconstructing climate trends. We also identify a few challenges that should be addressed in the future to further enhance GCM-based seasonal $ET_o$ forecasting.





Correction of lead-time-dependent errors should be further investigated in future GCM-based $ET_o$ forecasting. We found sharp

declines in the skill of calibrated $ET_o$ forecasts from lead time month 0 to month 1. Model initialization with field observations plays a critical role in seasonal climate forecasting based on GCMs (Doblas-Reyes et al., 2013; Hazeleger et al., 2013). Short-lead-time forecasts are more skillful since they are closer to the observed state of the climate system than those at long lead times. At long lead times, the predictable signal is often much smaller than the intrinsic uncertainty of GCMs. As a result, skills of raw forecasts often decrease quickly in the first month (Swapna et al., 2015), posing a challenge to statistical

calibration, even for those using sophisticated calibration models (Hawthorne et al., 2013). Currently, we calibrate raw $ET_o$ forecasts of each lead time independently. Whether correcting the lead-time-dependent biases will add extra skill to calibrated forecasts, particularly to those at long lead times, warrants further investigation (Schaeybroeck and Vannitsem, 2018).

Future forecast calibration should also investigate the impacts of climate change on the temporal variations of $ET_o$. In addition to the increasing or decreasing trends, warming climate also induced more significant temporal variations in $ET_o$, following

increasing climate extremes (Wen et al., 2012). The increasing variations could pose another challenge to statistical calibration models assuming an unchanged variance of observations. This current investigation provides a remedy for dealing with the varying mean of $ET_o$ in statistical calibration. Future investigations should evaluate whether allowing the variance to vary with time in calibration models would further improve the skills of seasonal $ET_o$ forecasts.

## 5 Conclusions

$ET_o$ forecasting provides useful information for hydrological investigations and has been increasingly used to support water resource forecasting and management. Anthropogenic disturbances have induced changes in the climate system and resulted in trends in many climate variables. GCMs often misrepresent these climate trends and thus lead to time-dependent errors in seasonal climate forecasts. We have recently improved the BJP model to deal with this error source through the reconstruction of observed climate trends in calibrated forecasts. In this study, we apply the BJP-ti model to calibrate raw seasonal $ET_o$

forecasts constructed with climate forecasts from the ECMWF SEAS5 model. The BJP-ti model effectively corrects misrepresented climate trends and reconstructs observed trends in calibrated $ET_o$ forecasts. More importantly, forecast skills in areas showing statistically significant observed trends in observations are improved following trend reconstruction. This investigation highlights the necessity of correcting time-dependent errors for enhancing GCM-based seasonal $ET_o$ forecasting. We conclude that future $ET_o$ forecasting based on GCM climate forecasts could obtain more skillful forecasts through climate

trend reconstruction.

This investigation also provides valuable insights for improving statistical calibrations of seasonal climate forecasts in the future. In recent decades, climate trends have been increasingly observed. However, many calibration models for seasonal forecasts have not taken the non-stationary behavior of the climate system into consideration. Improved forecast skills in seasonal $ET_o$ forecasts through the reconstruction of temporal trends, together with our previous calibration of seasonal





temperature forecasts, validate the robustness and effectiveness of bias-reconstruction algorithms in the BJP-ti model. We anticipate that these algorithms would be applicable to enhance other calibration models.

**Data availability:**

Data used in this study are available by contacting the corresponding author.

**Author contributions:**

Q. Yang and Q. J. Wang conceived this study. Q. J. Wang developed the calibration model. Q. Yang took the lead in writing and improving the manuscript. All co-authors, including A. Western, W. Wu, Y. Shao, and K. Hakala, contributed to discussing the results and improving the manuscript.

**Competing interests:**

The authors declare that there is no conflict of interest regarding the publication of this article.

**Acknowledgments:**

This study has been supported by an ARC Linkage Project (LP170100922). We thank the European Centre for Medium-Range
Weather Forecasts (ECMWF) for providing the SEAS5 data (https://www.ecmwf.int/). Computations of this research were undertaken with the assistance of resources and services from the National Computational Infrastructure (NCI), which is supported by the Australian Government. This research was supported by the Sustaining and strengthening merit-based access to National Computational Infrastructure (NCI) LIEF Grant (LE190100021) and facilitated by The University of Melbourne. The authors declare that there is no conflict of interest regarding the publication of this article.

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
