# Peer review of "Reconstructing climate trends adds skills to seasonal reference crop evapotranspiration forecasting"

_Hydrology and Earth System Sciences, 2021_

## Author Comment (AC1)

**Responses to Reviewer #1**

**Point #1**

*In the manuscript "Reconstructing climate trends adds skills to seasonal reference crop evapotranspiration forecasting", Yang et al adopted a new method to improve the prediction of evaporative water loss based on seasonal climate forecasts from the ECMWF model. This method is capable of dealing with the impacts of the changing climate on the prediction of future evapotranspiration (Reference crop evapotranspiration, ETo), and could lead to more realistic predictions. The changing climate has substantially altered the water cycle, representing one of the most critical challenges in hydrological modelling and water resource management. This work is innovative in taking this impact into account and addressing the challenges associated with climate change in the prediction of future evapotranspiration. The developed method is expected to be applicable to other models and thus benefit both forecasters (weather/climate centers) and forecast users (irrigators, hydrological modelers).*

*The manuscript is generally well written. Introduction clearly explains the background, challenges, motivation, and objective of this work; Method provides detailed information of the model, how the model runs are conducted, and evaluation metrics; Results generally are clear and readable; Discussion provides valuable insights and important implications for future improvements of climatology-based models in hydrological modeling and forecasting.*

*I encourage the authors to address the following issues before publishing this work.*

**Response: We appreciate the reviewer's nice summary and constructive comments.**

**Point #2**

*1. For time-series data, in addition to the magnitude of trend, another important feature is the statistical significance. I noticed the authors had taken this into consideration in selecting the months (8,9,10) for evaluating the performance of trend construction. In constructing the observed trends in calibrated forecasts, you empirically set limits of the trends in equation 8. I understand this is to avoid extremely large trend values. In addition to this adjustment, I think you should limit trends to zero, in grid cells where observed trends are insignificant (P<0.05). Otherwise, the trend reconstruction may overestimate climate trends. I see decreases in the correlation coefficients and skill scores when compared with the calibration without trend reconstruction (Figures 2 and 3). I think limiting the insignificant trends could avoid these unwanted decreases. I suggest the authors rerun the trend-reconstruction calibration and take statistical significance into account. If you see improvements in the new runs, update the results accordingly.*

**Response: We agree with the reviewer that the statistical significance of trends in observations should be tested and used to limit the reconstructed trends. We accepted your valuable suggestions and redid the calibration and analysis by setting limits in trend**

**reconstruction. Specifically, we used *P*<0.05 as the threshold to define statistically significant**

**trends. For grid cells with insignificant observed trends (*P*>0.05), we set inferred trends to**

**zero to avoid overfitting. We introduced this new strategy in section 2.3 as follows:**

"For trends that are insignificant (*P*>0.05), we set $m_i$ to 0 to avoid overfitting trends in calibrated forecasts. For significant trends, we set the $m_i$ value based on trends in observations and raw forecasts during 1981-2019"

**New results show that this strategy is not only effective in limiting the trend reconstruction to**

**regions where observed trends are significant, but also helps avoid the reductions in**

**correlation coefficient and CRPS skill score caused by overfitting (Figures 2 and 3):**

[Figure]

**Figure 2. Differences in the correlation coefficient (*r*) between BJP-ti calibrated forecasts and**

**observations with that between BJP calibrated forecasts and observations for three selected months**

**(AUG, SEP, OCT) and three lead times (Months 0, 3, and 6). Red polygons show regions with significant**

**trends.**

[Figure]

AUG     SEP     OCT

Month 0

Month 3

Month 6

Differences in CRPS skill score (%)

**Figure 3. Differences in CRPS skill score between BJP-ti calibrated forecasts and the BJP calibrated**
**forecasts for three selected months (AUG, SEP, OCT) and three lead times (Months 0, 3, and 6). Red**
**polygons show regions with significant observed trends.**

**We updated all results in the manuscript based on the new calibration.**

**Point #3**

*2. In addition to the improvements in the 3 selected months, whether trend construction improve the*
*calibration over the whole study period?*

**Response:  Thank you for the valuable suggestions. We added a new figure (Figure 4) to show**
**the overall improvements in CRPS skill score and updated section 3.3 accordingly:**

[Figure]

Differences in CRPS skill score (%)

**Figure 4. Differences in CRPS skill score between BJP-ti calibrated forecasts and the BJP calibrated**
**forecasts over 1990-2019**

Point #4

*3. Presentation of the improvements in figures 2 and 3. I suggest the authors use the percentage of*
*changes to demonstrate the differences. Since correlation and skill score vary largely from short to long*
*lead times, using percentages could better demonstrate the more significant improvements at long lead*
*times.*

**Response: Thank you for the valuable suggestions. We did not use percentage as the unit**
**because we found that at long lead times, CRPS skill score in calibrated forecasts based on**
**the BJP model could be slightly negative, and thus make the plot based on percentage**
**confusing:**

[Figure]

**As a result, we decided to use their original unit. Actually, after fixing the problems in**
**overfitting, figure 2 and 3 could better demonstrate how trend reconstruction improve the**
**correlation and skill scores, particularly at long lead times. Please see details in our response**
**to your comment #2.**

Point #5
*Specific comments:*

*Page 1. line 22, forecast should be forecasting*

**Response: We changed the wording accordingly.**

**Point #6**

*Page 3. line 92-93.  This study is performed across Australia only*

**Response: We add the following sentence to clarify the spatial extent of this investigation:**

"While SEAS5 produces climate forecasts across the globe, the calibration in this study is performed across Australia only."

**Point #7**

*Page 4. line 100, Calculation of ETo observations and forecasts*

**Response: We change the subtitle accordingly.**

**Point #8**

*Page 6. line 160-165. Please italicize k in this paragraph and throughout the manuscript to be consistent with the equations.*

**Response: We italicized *k* in the manuscript.**

**Point #9**

*Page 15. Figure 7, It is hard to read the alpha index values in the figure. Please consider changing the limits of the color bar, and use narrower limits (e.g.,0.8-1), to make the alpha index maps more readable.*

**Response:  We replotted the figure with a new color bar of 0.95-1 and replaced the original figure:**

[Figure]

**With the following one:**

[Figure]

**Point #10**

*Page 17. line 378.  To change with time?*

**Response:  We changed the wording based on your suggestions.**

---

## Author Comment (AC2)

**Responses to Reviewer #2**

**Point #1**

*1. General comments*

*This paper presents a method to improve monthly seasonal forecasts of potential evapotranspiration using a trend-aware statistical model (BJP-ti). This model builds on previous work by the authors on the BJP model combining a data transform with a multivariate normal distribution.*

*The topic of trend-aware forecasts is fundamental in a changing climate where the use of long historical time series to calibrate statistical forecast and post-processing models becomes questionable. This paper provides a valuable contribution to the field by showing how an existing statistical model can be extended to include trends with limited additional complexity. The model performance is thoroughly analysed using well established metrics that target a wide range of forecast attributes. Finally, the paper is well written, clear and to the point concise, with figures that provide strong visual evidence to support the authors' analysis.*

*We do not see any major issues with the paper and recommend it to be published with minor revisions. The two main items that could be improved by the authors aside of the detailed points raised in the following section relate to:*

**Response: We appreciate the excellent summary and assessment. We addressed the valuable comments carefully and provided point-by-point responses. Please see details as follows.**

**Point #2**

*[cross validation scheme] The authors used a traditional leave-one-out cross validation scheme where a single month is left aside for validation and the model is calibrated against the remaining data points. This an optimistic cross-validation scheme because the validation month is likely to show a similar trend and is not completely independent from the calibration data. A more conservative approach would be to split the data set in two parts, although this would not solve the problem completely. This an important issue but would require complex theoretical developments that are probably beyond the scope of this paper. However, we recommend a bit of discussion around this point.*

**Response: Thank you for the valuable comments and suggestions. We agree with the reviewer that the current leave-one-out cross-validation strategy is not perfect for inferring the trend parameters. As we introduced in the Method section (equation 5), the two trend parameters are inferred together with parameters (mean vector and covariance matrix) defining the bivariate distribution. The current strategy (leave-one-out) has been proven effective for the inferencing mean vector and covariance matrix, but may not be good enough for the inference of trend parameters, since the left-out month may not be fully independent of the remaining 29 months. Leaving out longer years, such as splitting the 30-year data**

**36**   **equally into two parts, as the reviewer suggested, could alleviate this problem to some**
**37**   **extent. However, we have another concern about data splitting. This strategy will**
**38**   **substantially reduce samples used for parameter inference from 29 to 15, and thus may lead**
**39**   **to significant sampling problems. We feel solving this problem may need additional efforts**
**40**   **and more sophisticated solutions. As a result, we highlight this as a challenge that should be**
**41**   **addressed in our future work in section 4.3 (Future work)**

**42**   "First of all, more sophisticated cross-validation methods should be developed for the inference
**43**   of trend parameters. The current leave-one-out method has been proven to be effective in the
**44**   inference of the mean vector and covariance matrix (Shao et al., 2020). However, this strategy
**45**   may not guarantee the independence between the left-out data and data used for the inference of
**46**   trend parameters. We decided not to implement the data-splitting method for cross-validation
**47**   because of the risk of introducing sampling errors. Future investigations should take this
**48**   challenge into consideration and develop better cross-validation methods for the inference of
**49**   trend parameters."

**50**

**51**   ## Point #3
**52**   *[risk of overfitting when there is no observed trend] The authors demonstrate that the BJP-ti model*
**53**   *outperforms BJP and raw forecasts when there is a trend in observed data. However, some of the results*
**54**   *shown by the authors suggest that its performance is worse than BJP when the trends are not significant.*
**55**   *This result is to be expected because of the higher number of parameters of BJP-ti which may increase*
**56**   *the risk of overfitting and counter-performance over validation data. We recommend highlighting this*
**57**   *point in the manuscript to better identify the strengths and weaknesses of BJP-ti.*

**58**   **Response: We agree with the reviewer that the original BJP-ti parameterization suffered from**
**59**   **parameter overfitting and resulted in degradation in performance when compared with the**
**60**   **BJP model.**

**61**   **To solve this problem, we accept your valuable suggestions and add limits to inferred trends**
**62**   **in trend reconstruction. Specifically, we use $P<0.05$ as the threshold to define statistically**
**63**   **significant trends. For trends that are statistically insignificant ($P>0.05$), we set the inferred**
**64**   **trends to zero to avoid overfitting:**

**65**   "For trends that are insignificant *(P>0.05)*, we set $m_i$ to 0 to avoid overfitting trends in calibrated
**66**   forecasts. For significant trends, we set the $m_i$ value based on observations and raw forecasts
**67**   during 1981-2019"

**68**   **This new strategy is not only effective in limiting the trend reconstruction to regions with**
**69**   **significant observed trends (Figure 1), but also avoids the reductions in correlation**
**70**   **coefficients (Figure 2) and CRPS skill score (Figure 3) following trend reconstruction.**

**71**   **We have updated all results based on the new calibration. We present Figures 2 and 3 here to**
**72**   **show the advantage and effectiveness of the new strategy.**

[Figure]

**Figure 2. Differences in the correlation coefficient (*r*) between BJP-ti calibrated forecasts and observations with that between BJP calibrated forecasts and observations for three selected months (AUG, SEP, OCT) and three lead times (Months 0, 3, and 6). Red polygons show regions with significant trends.**

[Figure]

**Figure 3. Differences in CRPS skill score between BJP-ti calibrated forecasts and the BJP calibrated forecasts for three selected months (AUG, SEP, OCT) and three lead times (Months 0, 3, and 6). Red polygons show regions with significant observed trends.**

 Point #4

*[Line 26] "Reference crop evapotranspiration (ETo) measures the evaporative demand of the*
*atmosphere": Please provide additional details regarding the definition of ETo. We suggest the following:*
*"Reference crop evapotranspiration (ETo) measures the evaporative demand of the atmosphere for a*
*hypothetical crop of given height, with defined surface resistance factor and albedo. It is generally*
*computed using the Penman-Monteith equation following Allen et al. (1998, see section 2.1), which is*
*known as FAO56. McMahon et al. (2013) provides additional information about the process. "*

**Response: Thank you for your valuable suggestions. We add the suggested introduction of $ET_o$**
**and the reference to the manuscript.**

**Reference:**

McMahon T.A., Peel, M. C., Lowe, L., Srikanthan, R. and McVicar, T.R.: Estimating actual, potential,
reference crop and pan evaporation using standard meteorological data: A pragmatic synthesis. Hydrol.
Earth Syst. Sci., 17, 1331–1363, doi: /10.5194/hess-17-1331-2013, 2013

Point #5

*[Line 94] "we combine the archived re-forecasts and operational forecasts": Please comment briefly on*
*the potential differences in skill between the re-forecast and operational data aside of the number of*
*ensembles generated.*

**Response: Thank you for the suggestions. According to the ECMWF SEAS5 documentations**
**(Stockdale et al., 2017; Johnson et al., 2019), SEAS5 runs for the re-forecast and operational**
**forecasts periods were configured as similar as possible to maintain consistencies. However,**
**there are some slight differences. In addition to ensemble size, initial conditions for the two**
**sets of runs are from different data sources. As a result, performance during the two periods**
**may vary for some weather variables. For example, according to the ECMWF user guide**
**(ECMWF 2021), because of the different initializations, '***the real-time forecasts of Lake*
*Superior (including the Great Lakes and the Caspian Sea) are cooler in the summer than the re-*
*forecasts were***'. In addition, according to the latest evaluation of the SEAS5 forecasts (Figure**
**40 in Haiden et al., 2021), forecasts of accumulated cyclone energy for the Atlantic tropical**
**storm demonstrate larger errors during 2016-2021 than the re-forecasts.**

**However, we feel it is hard to draw a conclusion on the relative performance of the re-**
**forecasts and operational forecasts, because they have different lengths and cover different**
**years, and their performances may vary with the ECMWF output variables.**

**In addition, we did not see significant differences in absolute errors in raw $ET_o$ forecasts**
**during the re-forecast period (1990-2016) vs. operational forecasts (2017-2019). We**
**calculated the average absolute errors in raw $ET_o$ forecasts across Australia during the study**
**period (1990-2019). The absolute errors during the re-forecasts and real-time periods seem to**
**be comparable. We added the following figure to the Supplementary material.**

[Figure]

Figure S1. Absolute errors in raw ECMWF ETo forecasts.

**123  Based on these investigations, we modified the introduction of the re-forecast and**
**124  operational forecasts as follows:**

"To match $ET_o$ observations, we combine the archived re-forecasts and operational forecasts to derive
raw $ET_o$ forecasts for the period of 1990-2019. ECMWF runs for the two sets of forecasts are configured
in a similar way, except for differences in initialization (Johnson et al., 2019). Absolute errors in raw $ET_o$
forecasts during the two periods are comparable (Figure S1). We choose the first 25 ensemble members
of the real-time forecasts (2017-2019) to match the ensemble size of the re-forecasts (1990-2016)."

**131  Reference:**

Stockdale, T., Johnson, S., Ferranti, L., Balmaseda, M. and Briceag, S.: ECMWF 's new long-range
forecasting system SEAS5. Meteorology section of ECMWF Newsletter No. 154., 2017.

Johnson, S. J., Stockdale, T. N., Ferranti, L., Balmaseda, M. A., Molteni, F., Magnusson, L., Tietsche, S.,
Decremer, D., Weisheimer, A., Balsamo, G., Keeley, S. P. E., Mogensen, K., Zuo, H. and Monge-sanz, B.
M.: SEAS5 : the new ECMWF seasonal forecast system, Geosci. Model Dev., 12, 1087–1117, 2019.

ECMWF. SEAS5 user guide. Version 1.2, March 2021.
https://www.ecmwf.int/sites/default/files/medialibrary/2017-10/System5_guide.pdf

Haiden, T., Janousek, M., Vitart, F., Ben-Bouallegue Z., Ferranti, L. and Prates, F.: Evaluation of
ECMWF forecasts, including the 2021 upgrade. Technical Memo 884. 2021.
https://www.ecmwf.int/sites/default/files/elibrary/2021/20142-evaluation-ecmwf-forecasts-including-
2021-upgrade.pdf

**144  Point #6**

*[Line 125] "trends in transformed forecasts and observations are removed to produce detrended data":*
*This is quite an aggressive process because removing trend linearly in transform space, as described in*
*equations 3 and 4, can lead to substantial reduction in un-transformed space after a certain time. When*

*trends parameters in BJP-Tri are significant (which seems frequent as suggested by Figure 1), we are a bit*
*concerned that this could lead to forecasts becoming unrealistically large or systematically zero if left*
*unchecked.*

**Response: We appreciate the reviewer's valuable comments. We further evaluated our**
**methodology results and confirmed that parameter inference in the transformed space did**
**not result in extreme values in calibrated forecasts. First of all, the removed trend will be**
**added back to transformed forecasts/observation through the retrending process (step 5 in**
**section 2.3). As a result, even a large trend is removed from transformed data in the**
**detrending process, it will be added back to the transformed data before calibrated forecasts**
**are transformed back to their original space. Second, as we introduced in section**
**2.3(equations 7 and 8), we've set limits to inferred trends to avoid extreme values. Third, we**
**further compared the absolute errors in calibrated forecasts produced using the BJP-ti model**
**vs. those using the BJP model (See the following figure), and did not see significant increases**
**in errors after trend reconstruction:**

[Figure]

Figure S2. Differences in absolute bias between BJP-ti and BJP calibrated forecasts

**The above figure indicates that differences in the two sets of calibrated forecasts (with vs. without trend reconstruction) are almost negligible. We added the above figure to the Supplementary Material, and explained findings in the comparison in section 2.3:**

"Our analysis indicated that our trend-reconstruction strategy (detrending and retrending in the transformed space, and setting limits to inferred trends) would not introduce unwanted bias in the calibrated forecasts (Figure S2)."

**As a result, we can reassure the reviewer that our trend reconstruction strategy will not lead to extreme values in calibrated forecasts.**

**Point #7**

*We suggest commenting briefly on the time needed for the mean unconditional forecast (i.e. considering*
*zo only in Equation 5) to depart from the unconditional forecast mean obtained at t=tm by more than,*
*say, 50% in untransformed space. Perhaps consider showing the distribution of this time across the*
*gridded domain and provide guidance on how frequently BJP-tri should be reviewed to monitor the*
*accuracy.*

**Response: Thank you for the comments. We create figures to show the time needed for the**
**departure of climatology forecasts which does not consider temporal trends from the**
**calibrated forecasts with reconstructed trends. Here we considered both 10% and 50%**
**departure. As we explained in our response to your comment #3, we adopted a new strategy**
**that only allows trend reconstruction in regions with significant observed trends. As a result,**
**we only focus on these regions when investigating the departures.**

[Figure]

Figure S11. Years needed for the departures of climatology forecasts from the calibrated
forecasts with reconstructed trends to exceed 10%

[Figure]

Figure S12. Years needed for the departures of climatology forecasts from the calibrated forecasts with reconstructed trends to exceed 50%

**As suggested by the above plots, it will take about 20-30 years for the departure to reach 10%, and more than 100 years to reach 50%. However, we believe correcting time-dependent errors is still necessary, since increasing extreme weather conditions across the globe in recent years indicate that climate change is intensifying. We add the following discussions to section 4.1:**

"Although it may take decades for climate change to substantially alter the magnitude of $ET_o$ (Figures S11 and 12), we recommend that future GCM-based $ET_o$ forecasting should still correct time-dependent errors. More frequent extreme weather events in recent years support model projections that climate change will intensify in the future (Kharin et al., 2013). It is expected that future climate change may induce more significant temporal trends in $ET_o$."

**Point #8**

*[Line 132] "t $\tilde{o}$ 🔲🔲🔲 is approximately the middle year": does moving tm has an impact on generated*
*forecasts? I believe not because it is compensated by the value of the mean parameter mu. Please*
*confirm. If this the case, please highlight that the position of tm is arbitrary and does not affect the*
*forecasts.*

**Response: Thank you for the valuable comments. The reviewer is right that using different**
**years as the reference for trend removal will impact the magnitude of the resultant**
**detrended data (both forecasts and observations), but will not affect the trend**
**reconstruction. When using a different year other than 2004 as a reference year, all**
**detrended data poinst will be larger (or smaller) by the same value than data using the**
**middle year as the reference. These differences will be lead to different mean and standard**
**deviation parameters. However, after we add the trend back (retrending) to data, the**
**difference will be canceled out. As a result, choosing a different reference year will not affect**
**the trend reconstruction and forecast calibration.**

**We clarify this point by adding the following explanations:**

**"The position of $t_m$ is empirically selected, but it will not affect the calibration if we choose a different**
**year as $t_m$"**

**Point #9**

*"Equation 8 shows the conditional posterior distribution of parameter $\tilde{o}$ 🔲🔲¼$\tilde{o}$🔲🔲🔲.": We suggest*
*"Equation 8 shows the posterior distribution of parameter $\tilde{o}$ 🔲🔲¼$\tilde{o}$🔲🔲🔲 conditional on $\tilde{o}$ 🔲🔲🔲🔲$\tilde{o}$🔲🔲🔲".*

**Response: We changed the wording accordingly.**

**Point #10**

*"In equation 8, $\tilde{o}$ 🔲🔲🔲🔲$\tilde{o}$🔲🔲🔲 is the mean and $\tilde{o}$🔲🔲🔲🔲$\tilde{o}$🔲🔲🔲 is the standard deviation for predictors or*
*predictands.": Please move this sentence just after Equation 8. In addition, we suggest the following*
*clarification: "$\tilde{o}$ 🔲🔲🔲🔲$\tilde{o}$🔲🔲🔲 is the standard deviation for predictors or predictands extracted from the*
*diagonal of covariance matrix S (see equation 5)".*

**Response: We moved this sentence to the beginning of this paragraph to better introduce**
**Equation 8. We also improved the descriptions of parameters based on your suggestions.**

**Point #11**

*[Line 160] "we adopt a leave-one-year-out cross-validation strategy": for a trend-aware model, this is an*
*optimistic approach to model validation because the model has seen both past and future data during*
*calibration. A more challenging validation would be to split the data in two parts, infer the trend from*

*one part and validate on the other. We understand that this is challenging with a heavily parameterised*
*model such a BJP, consequently it is probably beyond the scope of this paper to solve this question here.*
*However, it is important to flag the potential issue of using traditional leave-out validation for trend*
*analysis.*

**Response: We agree with the reviewer about the issue in the leave-one-out cross-validation.**
**Please see our response to the same point in your comment #2.**

**Point #12**

*[Line 166] "The comparison is conducted for months with large areas of statistically significant (at the*
*95% confidence interval) temporal trends in observed ETo.": this approach is problematic because it does*
*not check the performance of the BJP-ti model when there is no observed trend. BJP-ti is more*
*parameterised than BJP, consequently it is always exposed to the risk of overfitting the data when there*
*is no trend, i.e. when trend parameters cannot be calibrated reliably. Please comment on this point and*
*justify why performance assessment excluded month with no significant observed trend.*

**Response: Thank you for the valuable comments. As we explained in our response to your**
**comment #3, we adopted a new strategy to deal with the overfitting problem. In the latest**
**calibration with this strategy, the degradations in CRPS skill score and correlation coefficients**
**caused by trend overfitting have been effectively corrected.**

**We add the evaluation results for the remaining 9 months to the supplementary material. As**
**we can see in the following figures, improvements in the two metrics mainly occurred to**
**regions with significant observed trends. For regions with insignificant observed trends,**
**changes in the metrics are generally negligible. We introduced how results are presented in**
**section 2.4 as follows:**

"We present results of the comparison in the main text for months (August, September, and October) with
large areas of statistically significant (at the 95% confidence interval) temporal trends in observed $ET_o$;
results for the remaining nine months are presented in the Supplementary Material."

[Figure]

[Figure]

**Point #13**

*[Line 197] "$\dot{x}(t_i)$ is raw or calibrated forecasts of ETo (mm month-1)": This is a deterministic metric, so we believe that x(t) is the mean of raw or calibrated forecast. Please clarify.*

**Response: Thank you for the suggestion. The reviewer is correct that for raw forecasts, they are calculated with the ensemble mean of each input variable (temperature, solar radiation, and vapor pressure), so they are deterministic; for calibrated forecasts, we used ensemble mean here to calculate the bias. We further explained the differences as follows:**

"Raw forecasts are deterministic since they are calculated based on the ensemble mean of each input variable. For calibrated forecasts, we use the ensemble mean to calculate bias. "

**Point #14**

*"Observed ETo shows increasing trends in many parts of Australia in the three selected months": There is a significant body of literature related to trends in evapotranspiration related to climate change (McVicar et al., 2012). Please comment briefly on how this statement relates to current research in the field.*

**Response: Thank you for the valuable suggestions. We reviewed a few classic publications on temporal trends of ETo based on the reviewer's suggestions (Donohue et al., 2010; McVicar et al., 2012). Because these investigations focus on a period (1981-2006) earlier than our investigation (1990-2019), the negative trends across Australia from their research were not observed in our study. We add the following contents to briefly introduce analyses of temporal trends in ETo in Australia.**

"Compared with findings from previous investigations, observed trends identified in this study also demonstrate significant spatial variability and varying magnitudes in different months (Donohue et al., 2010; McVicar et al., 2012). We found more positive trends in our study period (1990-2019) than the period of 1981-2006 (Donohue et al., 2010) "

**Reference:**

Donohue, R.J., McVicar, T.R. and Roderick, M.L.: Assessing the ability of potential evaporation formulations to capture the dynamics in evaporative demand within a changing climate, J. Hydrol., 386 (1–4), 186-197, doi: 10.1016/j.jhydrol.2010.03.020, 2010

McVicar, T.R., Roderick, M.L., Donohue, R.J., Li, L.T., Van Niel, T.G., Thomas, A., Grieser, J., Jhajharia, D., Himri, Y., Mahowald, N.M., Mescherskaya, A.V., Kruger, A.C., Rehman, S. and Dinpashoh, Y.: Global review and synthesis of trends in observed terrestrial near-surface wind speeds: Implications for evaporation, J. Hydrol., 416–417, 182-205, doi: 10.1016/j.jhydrol.2011.10.024, 2012

**Point #15**

*[Figure 1.] We suggest adding the standard deviation of annual ETo in the first column of figure 1 to highlight the significance of trend values. It is important to understand if the observed trends of 6 to 8 mm/decade reported below are large compared to climatological variance.*

**Response: Thank you for the valuable comments. We add the standard deviation to the figure. We present the standard deviation in the last column because it is easier to show the legend. In response to your comment #17, we also add contour lines to show regions with significant observed trends. Figure 1 (Month 0) and results for other lead times (Month 3 and 6) in the Supplementary Material were all updated:**

[Figure]

**Figure 1. Trends in raw forecasts, BJP calibrated forecasts, and BJP-ti calibrated forecasts at the lead time of month 0, and observed $ET_o$ in August, September, and October. Blue polygons show regions where observed trends are statistically significant. SD refers to standard deviation.**

[Figure]

Figure S2. Trends in raw forecasts, BJP calibrated forecasts, BJP-ti calibrated forecasts for Month 3, and observed ETo for three selected months. Blue polygons show regions where observed trends are statistically significant. SD refers to standard deviation.

[Figure]

Figure S3. Trends in raw forecasts, BJP calibrated forecasts, BJP-ti calibrated forecasts for Month 6, and observed $ET_o$ for three selected months. Blue polygons show regions where observed trends are statistically significant. SD refers to standard deviation.

**Point #16**

*"Slight decreases in r are also found in regions where the observed trends are not statistically significant.": This statement seems to support the comment made against line 166 suggesting that BJP-ti might suffer from over-parameterisation when observed trends are not significant. If confirmed, this is an important limitation of the model that should be highlighted more clearly.*

**Response: We agree with the reviewer on the overfitting issue. We have explained how we address this challenge in our response to your comment #3. Specifically, we have set fitted trends for regions where observed trends are statistically insignificant to zero. This new strategy successfully resolved the overfitting problem, and degradation in performance of calibration following trend reconstruction (BJP-ti vs. BJP) was also corrected. We have updated the manuscript based on the new calibration.**

**Point #17**

*[Figure 2.] We suggest adding in this figure a contour line showing the area where observed trend is not significant. This could help understand better the strength and weaknesses of BJP-ti.*

**Response: Thank you for the valuable suggestion. After we adopted a new calibration strategy, as we explained in our response to your comments #3 and #16, degradation in the performance of the calibration was removed. We use contour lines to show the boundaries of regions with significant observed trends in Figures 1, 2, and 3.**

**Please see details in our response to your comments #3 and #15.**

**Point #18**

*Please also report the proportion of the study area where CRPS of BJP-ti is greater than the one of BJP. From Figure 3, it seems that BJP-ti underperforms in large parts of the domain, even if the decrease remains limited.*

**Response: Thank you for the comments. After we resolve the overfitting issues, degradation in forecast skills is removed. Please see details in our response to your comment #3.**

**Point #19**

*"with CRPS skill scores lower than -25% in all grid cells": this comparison is informative, but a little bit biased because raw operational forecasts are generally post-processed using techniques such as quantile-quantile mapping. We believe it is useful to show that raw forecasts have serious deficiency to reproduce on-ground observations, but it is also important to highlight that these forecasts would not normally be used for direct estimation of ET0.*

**Response: Thank you for the valuable suggestion. We agree with the reviewer that simple**
**bias correction is often applied to raw seasonal climate forecasts. We adopted quantile**
**mapping to raw ETo forecasts before the calibration with the BJP-ti model. However, we**
**found that bias-corrected ETo forecasts still demonstrate low skills for lead times beyond the**
**Month 0:**

[Figure]

Figure S13, CRPS skill score of bias-corrected ETo forecasts

"We need to point out that simple bias-correction is often applied to raw ECMWF forecasts
before they are used. We applied quantile mapping to the raw $ET_o$ forecasts and were able to
improve skills in $ET_o$ forecasts (Figure S13). However, the bias-corrected forecasts still
demonstrate skills much worse than climatology forecasts, particularly at long lead times."

**In addition, since the primary objective of this investigation is to understand how trend reconstruction would affect forecast calibration, we decided to use the raw $ET_o$ forecasts for this current investigation because we are not clear how would the quantile mapping affect trends in ECMWF forecasts.**

**However, we totally agree with the reviewer that improving the raw forecasts of ECMWDF forecasts will be a very interesting point which needs further investigation. Trends in individual input variables (e.g., temperature, vapor pressure, and solar radiation) needed for $ET_o$ calculation have been reported by Donohue et al. (2010) and McVicar et al. (2012). It is not clear whether correcting bias and reconstructing trends in each of the input variables first, prior to calculating the raw $ET_o$ forecasts, will further enhance the $ET_o$ forecasts calibration. We highlight this point in our Future work section (4.2):**

"In this study, we directly use the raw forecasts of individual input variables (e.g., temperature, solar radiation, and vapor pressure) to construct the raw $ET_o$ forecasts. However, trends in these variables have been reported in previous investigations. Whether correcting errors including time-dependent errors in the raw forecasts of each input variable, will lead to more skillful calibrated $ET_o$ forecasts, warrants further investigation in the future"

**Reference:**

Donohue, R.J., McVicar, T.R. and Roderick, M.L.: Assessing the ability of potential evaporation formulations to capture the dynamics in evaporative demand within a changing climate, J. Hydrol., 386 (1–4), 186-197, doi: 10.1016/j.jhydrol.2010.03.020, 2010

McVicar, T.R., Roderick, M.L., Donohue, R.J., Li, L.T., Van Niel, T.G., Thomas, A., Grieser, J., Jhajharia, D., Himri, Y., Mahowald, N.M., Mescherskaya, A.V., Kruger, A.C., Rehman, S. and Dinpashoh, Y.: Global review and synthesis of trends in observed terrestrial near-surface wind speeds: Implications for evaporation, J. Hydrol., 416–417, 182-205, doi: 10.1016/j.jhydrol.2011.10.024, 2012

**Point #20**

*It would be perhaps more interesting to compare the correlation score between raw and BJP-ti forecasts, which discards some the known deficiencies of raw forecasts.*

**Response: Thank you for the valuable suggestions. We agree with the reviewer that the correlation coefficient could be less impacted by the systematic errors in raw ECMWF forecasts than other metrics. We calculated the correlation coefficients between raw/BJP-ti calibrated forecasts and observations. Because of the high seasonality in $ET_o$, both raw and calibrated forecasts demonstrate high correlations with observations:**

[Figure]

Correlation coefficients between (a) raw forecasts/(b) calibrated forecasts and observations.

**To demonstrate the improvements in correlation through the calibration with the BJP-ti**
**model, we compared the correlation coefficients between calibrated forecasts and**
**observation with those between raw forecasts and observation:**

[Figure]

**(c) improvements in correlation coefficient through the calibration with the BJP-ti model**

**Results show improvements in correlation coefficients for all lead times, particularly in**
**northern Australia, where raw forecasts demonstrate low correlations with observations.**

**Since the correlation plots for (a) raw and (b) calibrated forecasts are very similar, we decided**
**to keep (b) and (c) in the main text (Figure 8 in the revised manuscript) and present (a) in the**
**Supplementary Material (Figure S10).**

**We add the new section in the main text to demonstrate the evaluation of the performance**
**of calibration in improving correlation coefficients:**

**"3.5 Correlation between raw/calibrated forecasts and observations**

The calibration based on the BJP-ti model also improves the correlation coefficients between forecasts and observations. Raw forecasts are able to capture the high seasonality in $ET_o$ and thus demonstrate high correlation coefficients with observations (Figure S10). The $r$ values are generally over 0.9 across most parts of central and southern Australia. Lower $r$ values are mainly distributed in coastal regions of northern Australia. Calibration with the BJP-ti model further improved the representation of $ET_o$ temporal dynamics (Figure 8). The $r$ values for calibrated forecasts are over 0.9 in most parts of Australia. Improvements in $r$ are more pronounced in northern Australia, where raw forecasts show lower correlations with observations. "

**Point #21**

*Same comment than for Line 290.*

**Response: We understand the reviewer's concern about how we evaluate the raw forecasts. As we explained in our response to your comments #19 and #20, we further 1) apply bias-correction to raw forecasts, 2) highlight the necessity of improving individual input variables prior to the calculation of raw $ET_o$ forecasts, and 3) use the correlation coefficients as another evaluation metrics to show the performance of raw forecasts. Please see details in our response to your comments #19 and #20.**

**Point #22**

*"We recommend that future GCM-based ETo forecasting should correct time-dependent errors": this comment should be toned down to include the risk of model overfitting discussed previously in relation to lines 166 and 271.*

**Response: Thank you for the comments. First, as we explained in our response to your comment #3, the overfitting problem has been resolved by setting the trend to zero in calibration for grid cells where observations do not demonstrate statistically significant trends. Second, we agree with the reviewer that it is necessary to remind the audience of the importance of avoiding overfitting in forecast trend reconstruction.**

**We feel it is better to highlight the necessity of dealing with overfitting in the discussion of BJP-ti model's strengths. As a result, we add the following discussions to the second paragraph of section 4.2 (Implications for improving statistical calibration models):**

**"**The successful application to ETo forecasts confirms the robustness of trend reconstruction algorithms based on the data transformation, Bayesian inference, and using statistical significance of observed trends to deal with overfitting of trend parameters in the BJP-ti model. This study further demonstrates the feasibility for the general application of BJP-ti to different hydroclimate variables showing temporal trends. We also anticipate that the BJP-ti algorithms for trend reconstruction could be adopted by other calibration models to enhance seasonal forecast calibration.**"**

**Point #23**

*"Future work for seasonal ETo forecasting": We suggest adding the two challenges of model overfitting when there is no observed trend and validation of trend-aware forecast beyond leave-one-out approach.*

**Response: Since the overfitting issue has been resolved (response to comment #3), and we already highlighted the importance of dealing with this issue in section 4.2 (response to comment #22), we decided to emphasize the challenge in cross-validation only here. Our discussion on the limitations of the leave-one-month out strategy and future work needed to address this challenge are presented in our response to your comment #2.**

---

## Author Response (AR2)

**Responses to Editor**

**Point #1**

Dear Qichun and co-authors,

Based on your responses and the carefully revised manuscript, I am pleased to accept your paper for publication pending some minor technical correction.

**Response: We greatly appreciate the editor's efforts in handling our submission during this challenging time. We followed your suggestions and improved the manuscript accordingly.**

**Point #2**

Line 169: the second 0.49 should be changed to 0.52. In your Track Changes version, I see 0.5 replacing 0.49.

**Response: We changed the value to 0.52 (now in Line 161).**

**Point #3**

Line 228: t should be in italic

**Response: We italicized the letter *t* (now in Line 218).**

**Point #4**

Figure 5: needs to be referred in the main text. It seems you missed to mention Figure 5 somewhere in Lines 336-340.

**Response: We cited the figure (Figure 5) in the text (now in line 317).**

**Point #5**

Figure 8: caption can be rephrased. I suggest: (a) Correlation coefficients between calibrated forecasts and observations, and (b) improvements in correlation coefficients through the calibration with the BJP-ti model relative to that with the BJP model

**Response: We improved the figure caption based on your suggestions (now in lines 343-345).**

**Point #6**

Congratulations and thank you for contributing to HESS!
Yi He, HESS Editor

**Response: We really appreciate your time and constructive comments. The work has been significantly improved through the review process. Thank you again for giving us this chance to introduce our work to the HESS audience. We hope the methodology developed through this study and the implications derived from the discussions will contribute to enhancing future $ET_o$ forecasting!**